# A Stepwise Male Introduction Procedure to Prevent Inbreeding in Naturalistic Macaque Breeding Groups

**DOI:** 10.3390/ani11020545

**Published:** 2021-02-19

**Authors:** Astrid Rox, André H. van Vliet, Jan A. M. Langermans, Elisabeth H. M. Sterck, Annet L. Louwerse

**Affiliations:** 1Biomedical Primate Research Centre, Animal Science Department, 2288 GJ Rijswijk, The Netherlands; astrid.roxrox@gmail.com (A.R.); vliet@bprc.nl (A.H.v.V.); E.H.M.Sterck@uu.nl (E.H.M.S.); louwerse@bprc.nl (A.L.L.); 2Animal Behaviour & Cognition (Formerly Animal Ecology), Department of Biology, Faculty of Science, Utrecht University, 3508 TB Utrecht, The Netherlands; 3Department Population Health Sciences, Division Animals in Science and Society, Faculty of Veterinary Medicine, Utrecht University, 3584 CM Utrecht, The Netherlands

**Keywords:** husbandry, colony management, inbreeding avoidance, social housing, demographic dynamics

## Abstract

**Simple Summary:**

Housing of primates in groups increases animal welfare; however, this requires management to prevent inbreeding. To this end, males are introduced into captive macaque breeding groups, mimicking the natural migration patterns of these primates. However, such male introductions can be risky and unsuccessful. The procedure developed by the Biomedical Primate Research Centre (BPRC), Rijswijk, the Netherlands, to introduce male rhesus macaques (*Macaca mulatta*) into naturalistic social groups without a breeding male achieves relatively high success rates. Males are stepwise familiarized with and introduced to their new group, while all interactions between the new male and the resident females are closely monitored. Monitoring the behaviour of the resident females and their new male during all stages of the introduction provides crucial information as to whether or not it is safe to proceed. Flow diagrams identify which decisions must be made during the introduction period. The BPRC introduction procedure is widely applicable to primates with natural male migration. Following this procedure may improve the management of captive primate groups in any housing facility worldwide. Altogether, careful introduction management can minimize the risk associated with male introductions and enhance the welfare of captive primates.

**Abstract:**

Male introductions into captive primate breeding groups can be risky and unsuccessful. However, they are necessary to prevent inbreeding in naturalistic breeding groups. The procedure used to introduce new individuals may affect the success and influence the risks associated with group introductions. At the Biomedical Primate Research Centre (BPRC) in Rijswijk, the Netherlands, male rhesus macaque (*Macaca mulatta*) introductions into naturalistic social groups with a matrilineal structure and without a breeding male achieve relatively high success rates. This paper describes the male introduction procedure used at the BPRC. Males are stepwise familiarized with and introduced to their new group, while all interactions between the new male and the resident females are closely monitored. Monitoring the behaviour of the resident females and their new male during all stages of the introduction provides crucial information as to whether or not it is safe to proceed. The BPRC introduction procedure is widely applicable and may improve the management of captive primate groups in any housing facility worldwide. Thus, the careful introduction management can minimize the risk associated with male introductions and enhance the welfare of captive primates.

## 1. Introduction

Animals play an important role in the life of many people. Animals are important companions, and are also often used for education and conservation in zoos and research facilities. The welfare of all animals should be of major concern for everyone involved, including biomedical researchers involved in animal testing. Impaired animal welfare is associated with physiological and behavioural changes that may affect the outcome of biomedical and behavioural studies (e.g., [1,2,3,4]). Improving animal welfare in research facilities may therefore result in more reliable data and the reduction in and refinement of animal experimentation [5].

An important marker of animal welfare, commonly used by researchers assessing an animal’s quality of life, is the expression of natural species typical behaviour (e.g., [6,7,8]). The expression of natural social behaviour is important for animals living in social groups, particularly primates. The welfare of captive primates can be impaired when the breadth of the displayed social behaviour repertoire is limited due to housing conditions. Primate welfare can be improved when they have the opportunity to engage in complex social interactions, similar to the interactions observed in wild primate groups (e.g., [9]). Therefore, it is important to house primates socially, preferably in naturalistic social groups where they can display natural social interaction.

Many research institutes worldwide still house their primates under non-naturalistic circumstances, such as pair-housing or peer groups. While such groups allow social behaviour and are preferred over solitary housing, group composition and migration patterns differ from wild groups. In contrast, the primates in the breeding colony of the Biomedical Primate Research Centre (BPRC), mainly macaques (*Macaca spp.*), are housed in naturalistic social groups that typically contain between 10 and 30 individuals. The composition and migration patterns within these groups is managed so that they closely resemble those in nature and allow the expression of natural complex social interactions. Generally, wild macaques live in multi-male multi-female groups [10], but one-male groups have also been observed [11,12,13,14,15,16]. Similarly, the groups at the BPRC consist of multiple adult females and their dependent offspring. The groups typically contain one adult breeding male, since early efforts to introduce multiple males were not successful. Females are philopatric and remain in their natal group throughout their lives, similar to wild groups [17,18,19,20,21,22,23]. Consequently, these captive females form matrilines (i.e., families related in the female line), which are important for increasing group stability [24,25]. Males are removed from their natal group at puberty, i.e., at an age of 2.5 to 5 years; the same age at which wild macaque males would leave their natal group [17,19,22,23,26,27]. However, when female offspring remain in their natal group, the risk of inbreeding between a breeding male and his maturing daughters arises. Therefore, breeding males need to be replaced every three to five years to prevent inbreeding with maturing daughters, similar to wild male macaques, who change breeding groups several times in their lives [17,18,22,23].

The introduction of a new breeding male into an established social group may be risky. New males may be rejected by the females and introductions may be unsuccessful [25,28]. In extreme cases, females may injure or even kill new males [25]. Females and their offspring are also at risk during male introduction. Females may become injured in fights with new males [25,29], and new males may kill dependent offspring (e.g., [30]). Moreover, new males and resident females may experience high stress levels during introductions, similar to male group entry in the wild [31,32,33]. Due to these risks, it is important to consider all management strategies to facilitate social group formation and to prevent inbreeding. One option is to remove all offspring before maturation. The current age-norm for removing macaque offspring from their natal group is 10 to 14 months [34]. Many research institutes adhere to this age norm and rear their macaques in peer-groups. However, peer-reared individuals experience high levels of stress early in life, which has long-term consequences on their behavioural and physiological development and may affect biomedical research results later in life [1,2,3,4,35]. Moreover, peer-reared individuals are often socially incompetent and form less stable groups [2,25,36]. Thus, removing offspring from their natal group before reaching adulthood compromises animal welfare and is not an ideal way to prevent inbreeding in primate groups. Naturalistic group housing with necessary male introductions can be a viable option to prevent inbreeding in captive primate groups.

The risks associated with male introductions may be minimized when the introductions are carefully planned and managed. At the BPRC, approximately 77% of the male introductions are successful (i.e., the male remained in the group for at least 4 weeks full time [25]), while 59% of all introductions lead to long-term stable groups (i.e., males reside in the group until they are removed for management reasons [25]). These high success rates are comparable to the success rates observed after pair formation in rhesus macaques (e.g., [37,38]), a far less socially complex situation. Similarly, data from take-overs by aggressive wild male macaques show that 3 out of 7 (57%) [16]; or 4 of 6 (67%) [23] and wild capuchin monkeys (2 of 11 (18%), [39] fail. Studies focussing on pair formation or the formation of new, non-naturalistic, social groups show that the success of an introduction may depend on the procedure used to compose the group [38,40]. However, there is little information regarding the introductions of new males into established breeding groups. Sharing the BPRC introduction procedure may therefore help other facilities that house captive macaque breeding groups, including zoos and research institutes, to improve their introduction management strategy. This allows the comparison of male introduction methods, the search for optimal introduction procedures and minimizing the risks associated with male introductions while enhancing animal welfare.

## 2. Methods

### 2.1. Housing Conditions

The BPRC houses two macaque species, rhesus macaques (*Macaca mulatta*) and long-tailed macaques (*M. fascicularis*). All macaques in the BPRC breeding colony are group-housed in inside (72 m^2^, 2.85 m high) and outside enclosures (208 m^2^, 3.1 m high). In total, the BPRC (summer 2019) houses 28 breeding groups of rhesus macaques and 12 breeding groups of long-tailed macaques. Animals have the opportunity to move freely between their inside and outside enclosures, during both day and night. Both enclosures contain a variety of environmental enrichment items, such as elevated beams, fire hoses, climbing structures and a pool, to stimulate natural behaviour [41].

The inside and outside enclosures contain different compartments, either separated by wire mesh or concrete walls (Figure 1). The inside enclosure consists of four different compartments, the outside enclosure of three different compartments. The different compartments always have two or more connections to the other compartments, preventing high ranking group members from monopolizing access to certain compartments. These different compartments are used to introduce a new male into the group stepwise and allows animal caretakers to regulate a new male’s access to the group during the introductions. 

The breeding groups consist of adult females of a varying age (>3 years) and their non-adult offspring. To mimic natural migration patterns, females remain in their natal group whereas the males are removed when reaching sexual maturity, i.e., between 2.5 and 5 years of age. These removed males are housed in small all-male groups for several years until they are old enough to become a breeding male and are introduced into a breeding group. When groups become too large and unstable, they are split into several new groups while maintaining the natural matrilineal structure of the group. This strategy leaves the matrilineal structure of the groups intact, which increases social stability [25,36].

### 2.2. Developing the BPRC Introduction Procedure

The BPRC introduction procedure was developed based on personal knowledge and experience, in particular those of A.H.v.V. and A.L.L. As there is only limited information on male introductions into breeding groups, this personal knowledge was used in this paper to explain particular decisions during the introduction process (referred to as personal communication).

### 2.3. Conducting Male Introductions at the BPRC

At the BPRC, adult breeding males are removed from their group when the risk of inbreeding with adult daughters arises, i.e., when they become 4–5 years old. The median length of a successful introduction is 18 days (*n* = 49). One successful introduction took 357 days [25]. Usually, such a long introduction period is only carried out in exceptional cases, in this case because the man was genetically important for this breeding group. A new male is introduced according to the BPRC introduction guidelines, wherein males are familiarized with and introduced into a new group stepwise. During the introductions, the new males need to obtain the alpha position immediately after group entry, mimicking bluff immigration in the wild [16]. If the introduced male is not dominant to all females, there is a risk that he might be severely injured by the females. The process of introducing a new breeding male is described below. There are five steps, all crucial for optimizing introduction success and long-term group stability, namely: (1) selecting a breeding male; (2) preparing the male and the group for the introduction; (3) familiarizing the new male and the group; (4) physically introducing the new male; and (5) post-introduction monitoring. 

The introductions are carried out by specialized caretakers under close observation by behavioural specialists. The caretakers decide on the duration of the different phases of the introduction, depending on the specific behaviour of the animals involved. The introduction process is focussed on rhesus macaques at the BPRC but is also used for long-tailed macaques. However, the timing of the introductions differs between these species. Rhesus macaque introductions take place between October and March, which corresponds to their breeding season, and is the natural timeframe for male dispersal in this species [17,19,28]. Long-tailed macaque introductions take place throughout the year, as long-tailed macaques are non-seasonal breeders where male dispersal occurs throughout the year [23].

### 2.4. Male Introduction Procedure

Step 1.Selecting a Breeding Male

Matching the right male to the right group is the first crucial step in increasing male introduction success. Biomedical research facilities often have multiple breeding groups and multiple breeding males available. At the BPRC, males are selected based on several parameters, of which genetics is one of the most important factors (step 1A in Figure 2). It is key to maintain genetic diversity, based on their major histocompatibility complex (MHC) [42,43], in captive populations and minimized inbreeding [44]. Males that are genetically overrepresented in the colony are excluded from introductions, while males with rare MHC genes in the colony are particularly preferred breeding males. Moreover, male genetics should be dissimilar from the resident females’ genetics [44]. This may also contribute to successful introductions, as females generally prefer genetically dissimilar males over males with genetics more similar to them (e.g., [45]).

After genetic matches are made, social behaviour and social experience are assessed (step 1B in Figure 2). First, it is important that the new male is reared in a social group and stays in his natal group for least 3.5 years to ensure proper social development [25]. Moreover, the selected male should not have caused social problems in his natal group (e.g., persistent aggression leading to veterinary treatment towards several animals). After removal from his natal group and being transferred to an all-male group, he should not be involved in extensive aggression and not become injured or injure other males that require veterinary treatment. Moreover, he should not be afraid of other animals or humans, engage in positive social interactions with his group members, and actively participate in positive reinforcement training. Usually, these males turn out the be males from middle-ranking matrilines in their natal group (personal communication). 

Finally, age and body condition are used to decide which of the candidate males will be introduced into the group (step 1C in Figure 2). Within the all-male groups, removed males have the opportunity to mature to full adult male. After obtaining adult male body features, they can be introduced to a breeding group from the age of 5. These relatively young and inexperienced breeding males are usually introduced into smaller breeding groups, i.e., typically below 15 individuals, to increase the chances of successfully obtaining the alpha position (personal communication, [16]). Young unexperienced males should not be introduced into groups with a history of extreme hostility against new males and unsuccessful male introductions. After they gained experience in their first breeding group and have reached prime age (i.e., between 9 and 12 years [25]), males may be introduced into larger and more complex social groups. These prime aged males are usually more successful during introductions [25,46].

Step 2.Preparing the Male and the Group for the Introduction

When a male is selected for introduction into a particular breeding group, the introduction is carefully planned and executed to increase the chances of success.

First, the residing alpha male needs to be removed from the group, possibly alongside the sub-adult natal males (i.e., males aged 3 or more) (step 2A in Figure 3). Male rhesus macaques should be removed before the start of the breeding season to prevent female pregnancy, since this decreases the chances of successful male introductions [25]. At the BPRC, residing males are preferably removed from the group at the start of summer, in June or July. Groups always spend at least four weeks without the previous breeding male in their group before a new male is introduced. This allows the group to settle in their new situation and stabilize without an adult male present in their group (personal communication). However, the removal of natal males is not necessary when a new male is introduced into the group right before or early in the breeding season, which corresponds with the natural timing of male group entry in the wild [17,19,28]. At the BPRC, natal males may breed with their (half)sisters or other (related) females in their group from the age of 2.5 [47]. The presence of natal males does not affect introduction success [25], and introducing a new male early in the breeding season may prevent the subordinate natal males from reproducing. However, when a group spends one or two breeding seasons without a breeding male, the removal of natal males is necessary to prevent inbreeding. Introductions into groups that spend at least one breeding season without an adult male may be beneficial, as there are no infants present in these groups. This practice may remove a trigger for escalating female–new male aggression, which often starts when infants squeak or scream towards new males (personal communication). Bernstein et al. reported that all male introductions into groups of lactating mothers outside the breeding season were unsuccessful [28]. However, data collected during 64 introductions at the BPRC rhesus macaque breeding colony show that the presence of infants does not affect introduction success [25]. Therefore, when high reproductive output is needed in the breeding colony, a new male can be introduced in the breeding season immediately following the removal of the previous breeding male.

Second, it is important to observe the group before a new male is introduced (step 2B in Figure 3). The females that may have a large influence on the group’s behaviour, such as the alpha female and the matriarchs, need to be identified. Moreover, it is necessary to determine the rank order of group matrilines and which females belong to those matrilines. This information is crucial during the introduction, as female dominance rank may help to predict whether fights between females and a new male are likely to escalate and help to identify the culprits (personal communication). Females within the same matriline are likely to support each other in conflicts [36,48,49,50]. It is also necessary to understand the matrilineal group structure when the introduction is unsuccessful and the group needs to be split for some reason (e.g., a matrilineal overthrow due to social instability, or certain females that continue to aggress their new male). When the group structure is known, the group can be split relatively quickly while leaving the natural matrilineal structure of the group intact. It is important to conduct the observations after the previous breeding male and the sub-adult natal males are removed and the females of the group have settled in their new situation, as the presence of the males may affect female behaviour.

Third, once the group structure is known and the important females have been identified, the new male can be moved to an enclosure adjacent to the females (compartment 1 in Figure 1, step 2C in Figure 3). In this housing condition, the male is separated from his future group through a concrete wall and is housed alone. This situation allows the male to become acquainted with the group and their neighbours through auditory, olfactory and limited visual contact, and recover from any potential stress associated with cage relocation (see [51]). This practice minimizes the chances of neighbouring groups interfering with the introduction (personal communication). Moreover, it is crucial for a male to understand that he is removed from his previous group and will not gain any support from previous group members during the introduction. This solitary period may prevent escalated aggression from the new male to the members of his new group (personal communication). At the BPRC, the male is housed adjacent to the females for approximately two weeks before the introduction proceeds. The importance of the new male and the neighbours becoming acquainted with the new situation always needs to be balanced against the male being housed solitarily, which may also impair his welfare. Therefore, it is important that the introduction proceeds as soon as the new male and the neighbours are accustomed to the new situation, based on their behaviour.

Step 3.Familiarizing the New Male and the Group

In the wild, new males may follow and observe groups from a distance before they attempt to enter or take over the group [23]. During this period, males may become gradually acquainted with the group and estimate the chances of successful group entry. A similar procedure, wherein a male is familiarized with his new group stepwise before physically introducing the male to the group (step 4), is used during male introductions at the BPRC. During the familiarization phase, the male is only able to interact with the group through wire mesh and is not yet physically introduced to the group.

The first step of the familiarization phase is moving the male to another compartment of the inside enclosure where he can see the group through wire mesh, but not yet touch them (compartment 3 in Figure 1, step 3A in Figure 4). During this phase, the group has access to its entire outside enclosure, and two inside compartments (compartments 1 and 2 in Figure 1). The male will remain in this compartment until the first aggression between him and the group diminishes, which usually occurs after one or two days (personal communication). Then, the male will get additional access to a compartment where he can physically interact with the group through wire mesh (compartment 4 in Figure 1, step 3B in Figure 4). Interaction through the mesh is always supervised and is only allowed for a limited period of time each day. Generally, there are high levels of aggression between the resident females and the new male during the first interactions through the wire mesh (personal communication). Therefore, interaction is only allowed for up to one hour on the first day. After this hour, the male is moved back to the compartment 3 (Figure 1), where he can see but not touch the females. On the second and third day, aggression levels usually decrease drastically and the first females may show interest in the new male and display affiliative or sexual behaviour through the wire mesh (personal communication). If aggression levels between the females and the new male remain high for several days and there are no signs of diminishing aggression, the introduction is stopped. This is particularly important when there is a risk of severe injuries on fingers, arms, and sometimes the face through the wire mesh. Additionally, the introduction is stopped when the new male displays submission by showing bared teeth display to any resident group members, the male is not interested in interacting with the females at all and stays away from them for several days, or when females do not show any positive interest (i.e., display mating or affiliative signals) in the new male. When particular females remain highly aggressive to the new male or start the majority of conflicts with the new male, these females may be individually introduced to the new male. In this case, the aggressive female is separated from the group and physically introduced to the new male in the inside enclosure, while the group is locked outside without visual access to the introduced pair. This visual barrier prevents the group from interfering in interactions between the new male and the female. Females may spend up to two hours with the new male. Individual females should not be separated from the group for too long (at most 4 h), as this may lead to social instability and the matrilineal overthrow within the group (unpublished data; [36]). Usually, aggression levels diminish after this one-to-one contact between highly aggressive females and their new male (personal communication). If one or a few females remain highly aggressive towards the new male even after one-to-one contact, the introduction of this male cannot continue in this group. Often, the introduction is stopped and considered unsuccessful.

However, sometimes, the female(s) aggressing the new male during an introduction have a history of repeated extreme aggression towards new males, and their behaviour has led to several unsuccessful introductions. In this case, one could consider permanently removing the repeatedly aggressive female(s) from the group. However, the removal of individual females from a group may cause social instability [36], and can only be done when the remaining members of the female’s matriline will be able to defend their rank without the presence of the removed female. Moreover, the welfare of the removed female will be impaired, as she will experience stress from moving [51] and will be alone after removal from her group. Therefore, this measure should only be taken in extreme circumstances, and requires careful deliberation about all possible options. An alternative strategy may be removing the entire matriline of the highly aggressive female(s) from the group, forming two different breeding groups.

When female–new male aggression decreases and female interest increases during the first three days of contact through the wire mesh and the situation remains stable for at least one week, the introduction can proceed. During this stage of the introduction, the male will gain additional access to one outside compartment, allowing him to interact with the females through the wire mesh in both the inside and outside enclosure. Moreover, he will have access to these compartments separating him and the females through wire mesh for 24 h/day (step 3C in Figure 4). Access to the outside compartment is important as this may promote exercise. At the BPRC, animals are more active outside than inside (unpublished data). Outside access may therefore enhance muscle formation, which in turn, may increase introduction success [25]. Moreover, males that are more fit can escape coalitionary attacks by females more easily if they should occur (personal communication). Therefore, it is important to ensure the new male is in good physical shape before he is physically introduced to his group. This introduction stage will continue for one to two weeks.

The final step of the familiarization phase depends on the new male’s social history: whether he is inexperienced or experienced. If the male is young and inexperienced, he will be physically introduced to small sub-groups of females that show affiliation to the male each day for approximately two weeks (step 3D in Figure 4). The females will spend up to two hours per day with the male in his compartments of the inside enclosure (compartments 3 and 4 in Figure 1), while the remainder of the group can see them through wire mesh (access to compartments 1 and 2, and their outside enclosure). During the remainder of the time, the male is able to interact with the entire group through wire mesh. Again, is it important to only separate females from their group for a limited amount of time each day to prevent social instability. This full physical access to part of the females may help an inexperienced male practice his social skills and mating behaviour as an alpha male (personal communication). After two weeks of various small sub-groups of females visiting the new male, the physical introduction may start. If the new male is an experienced breeding male, this final step with small sub-groups of females may be skipped and the physical introduction may start after the male had access to the females 24 h/day for one to two weeks through the wire mesh. Note that if aggression between the females and their new male flares up again at any time during the familiarization phase, the introduction reverts to an earlier step in the procedure to see whether aggression decreases again. If the aggression does not decrease, the introduction can be stopped at any time during this process.

Step 4.Physically Introducing the New Male

If the familiarization phase is successful, the male is physically introduced to the group. As there are no other adult males in the group, the new male should become the alpha male immediately after introduction into the group. Therefore, the introduced males can be seen as bluff immigrants, who attempt to obtain the alpha position right after group entry in the wild [16].

The physical introduction of a male starts in the outside enclosure, which consists of three different compartments separated through wire mesh. To increase the chances of the male successfully escaping a potential coalitionary attack by his new group, he gets the opportunity to explore the outside enclosure before he is introduced to the group (step 4A in Figure 5). The group will be locked inside, while the male gets access to his inside compartment and the entire outside enclosure for up to two hours. Usually, this is done the day before the physical introduction starts.

Then, on the day the male is physically introduced to the group for the first time (step 4B in Figure 5), the group will be locked outside, limiting their access to their two outside compartments (see Figure 1). The physical introduction is started in the outside compartments to allow the experienced animal caretakers conducting the introductions to oversee the entire enclosure. This is not possible when the animals have access to both their inside and outside enclosures. This overview is crucial at the start of the physical introduction to be able to identify which individuals cause problems, and quickly interfere with the introduction when necessary. The male, familiarized with the compartment next to the group, typically quickly returns to this compartment when aggressed by females, allowing easy separation. Moreover, the presence of an animal caretaker limits aggression during the introduction, as animals may be less inclined to impulsively attack each other in the presence of people (personal communication).

During the first step, the male is housed in the third outside compartment to allow interaction with the group through wire mesh, as described in the familiarization phase. If aggression levels are low, the sliding doors between the compartment will be opened. The male and the group will have access to the entire outside enclosure, for approximately 1–1.5 h on the first day. During the remainder of the day, the male will be able to interact with the group through wire mesh from his two inside (compartment 3 and 4 in Figure 1) and one outside compartment. The time a male spends time with the group each day is gradually increased. All interactions between him and his new group members are closely monitored by the specialized animal caretaker. Some females may tolerate the new male in their group from the first day, while others are more hostile [52]. Unwilling females may defend themselves when the new male approaches them, which may result in a coalitionary attack on the new male. Therefore, the male should wait for the females to approach him, and only take initiative for interactions after she approached him first (personal communication). When a male continues to approach the females who avoid him and run away from him, the introduction is stopped for that day and the male is returned to his own compartments where he can interact with the group through the wire mesh. If this behaviour continues for several days, the introduction is stopped.

The beginning of physical introductions is characterized by high levels of aggression, submissive behaviour by females to the new male and mating, regardless of who initiates these behaviours; in contrast, bouts of affiliation (e.g., grooming) are rare. Over time, aggression and submission rates should decline, while affiliation increases [28,52]. High levels of aggression at the start of the introduction are in principle not problematic, female threats to the male may even confirm this dominant position and show his strength and social skills by responding with appropriate counter aggression (personal communication). Generally, there is no need to interfere with aggression as animals will be able to sort out their new social position (personal communication). Intervention is only taken when aggression is extreme, i.e., there is risk of severe injury requiring veterinary treatment or of social defeat. Socially defeated males crouch passively and cease resistance to group attack [53]. Socially defeated males are likely to be killed by resident females when they are not removed from the group immediately (personal communication). When there is a substantial risk for social defeat of the male, the introduction is stopped and considered unsuccessful. Moreover, the male should never be seriously injured during the introduction, but female injuries may occur at the start of the introduction. If the new male becomes injured, the introduction must be stopped right away to prevent females from winning conflicts from the new male (personal communication). Within three days of the physical introduction, aggression levels should have decreased. If the group still responds with high aggression to their new male, the introduction is stopped. If only one or a few females remain highly aggressive, these females may be introduced to the new male alone, similar to the familiarization phase. In general, solo introduction can help decrease the hostility of highly aggressive females towards a new male. If not, one could consider to halt the introduction and remove the aggressive female(s) from the group, or to split the group (see above). The introduction should also be stopped if the male shows submission to or loses conflicts from resident animals to prevent social defeat (personal communication). Finally, the introduction should be stopped if the male repeatedly responds with aggression to non-threatening social situations, as this shows social incompetence [2]. In summary, the first days of an introduction must be carefully monitored as the behaviour displayed during this period shows whether the introduction has potential to succeed or should be stopped.

The introduction progresses when several females tolerate the new male in their group (see [52]) and no physical or serious aggression has been observed for at least one week. First, the animal caretaker’s supervision decreases (step 4C in Figure 5). The caretaker will move out of sight for some time when the male has physical access to the females but will remain within hearing distance from the group. Consequently, the behaviour of the animals is no longer affected by the presence of humans. Still, severe aggression will be noticed as this is generally accompanied by loud vocalisations (personal communication). Making use of video observations is helpful during this stage, as this will allow the identification of the initiator of a conflict. If there is no increase in aggression or decrease in female interest in the new male, the animals will gain additional access to their entire enclosure whenever the male spends time with the group. The time the male spends with the group gradually increases to approximately eight hours per day (i.e., a full workday) with minimal supervision (step 4D in Figure 5). When this goes well for at least one week, and at least half of the females in the group tolerated the new male and engage in positive social interactions (e.g., mating, proximity, or affiliation) with him, it is safe to leave the male in the group full time (personal communication). It is not necessary for all females to tolerate the new male in their group as their relationship may develop after the introduction [52]. However, the majority of the females need to tolerate the male, as this minimizes the risk of female coalitionary attacks on the new male (personal communication). The decision to proceed with the introduction is based on knowledge and experience. However, usually, when the male is completely settled in his new group, he becomes less cooperative with separation and returning to his own compartments at the end of the day. This is an additional sign that the male is ready to remain in the group full time (personal communication). To increase introduction success, it is crucial to take steps slowly, and give the animals time to establish new social relationships (personal communication). Thereby, not only aggression but also the establishment of positive social relationships between the females and their new male should be considered.

Step 5.Post-Introduction Monitoring

It is important to continue to monitor the group after the introduction has finished. Not all social relationships may have been established during the introduction. In addition, new contexts that the animals experience, such as feeding or sleeping arrangements at night, may trigger aggression. Therefore, the group and the male should be checked immediately in the morning following his first night in the group. Hereafter, they are extensively checked once a day for at least 30 min and regularly checked twice a day. Special attention is paid to possible injuries and other signs of social instability (e.g., increased stress-related behaviour or aggression). The introduction is considered successful when the male has stayed with the group full time for four weeks [52].

## 3. Discussion

This paper describes the procedure used to introduce an adult breeding male into a naturalistic group of rhesus macaques at the BPRC. Introducing new males is necessary to prevent inbreeding between maturing daughters and adult breeding males in naturalistic macaque groups. However, male introductions are risky and may not always result in success, aspects which can be influenced by the procedure itself [38,40,54]. To our knowledge, this paper is the first to describe a successful procedure for introducing an adult male into a naturalistic group of rhesus macaques in detail. The male introduction procedure used at the BPRC shows relatively high success rates: 77% were successful (i.e., the introduced males remained at least 4 weeks full time in the group), while 59% of all introductions led to long-term stable groups (i.e., males can reside several years in the group until they are removed for management reasons such as a risk of inbreeding) [25]. Sharing this procedure will help improve male introduction management at other facilities that house captive macaque breeding groups.

Key to the BPRC introduction procedure is the careful monitoring of the behaviour of the animals involved. Particular attention should be given to the male’s ability to win fights against female coalitions, as it is important that the male obtains and maintains the dominant position in the group. Once a male loses in a conflict with a group of females, which may result in social defeat, he will not be able to obtain a dominant position in another breeding group (personal communication). These observations are in line with the winner–loser effect, which is widespread throughout the animal kingdom and shows that once an individual loses a conflict, they are significantly less likely to win future conflicts (reviewed in: [55]). Therefore, it is crucial to introduce a new male stepwise and the introductions are conducted by people that are experts in primate behaviour. At the BPRC, specialized animal caretakers conduct the introductions, in close collaboration with a specialist in macaque behaviour. These caretakers attempt to minimize the risks of males losing conflicts from females by monitoring aggression, submission, and positive social interactions between females and their new male. Moreover, they take notice of who initiates conflicts and triggers of aggression to identify individuals that are a particular risk to the introduction. Whenever the caretakers are in doubt as to whether an introduction progresses well, they will not proceed to a next step of the introduction process, or even revert to an earlier step. Taking additional time does not harm the introduction, whereas introductions that proceed too fast can have detrimental consequences (personal communication). When an introduction takes too long, it is questionable whether the male and the group are a good match. In this case, stopping the introduction may be the preferred option. However, we have had one successful case with an introduction phase of almost 1 year.

We described the steps that can be taken during macaque male introductions and have identified potential behaviours signalling whether the introduction is progressing well or not, but it is impossible to cover all interactions during male introductions. It is important to realize that every introduction is different, and there is no perfect way to introduce a new male into a group. Small, unanticipated events, such as a squeaking baby, may already trigger a coalitionary female attack on a new male, even when the introduction progressed well for several weeks [25]. Therefore, flexibility in the process of introducing a new male is required, and the introduction should remain under close supervision.

There is only limited published information on macaque male introduction management. The few publications analysing the factors affecting the outcome of male introductions (e.g., [25,28]) are generally in line with the BPRC introduction procedure, which strengthens the notion that the BPRC introduction procedure is a relatively well-established method. Moreover, the success rate (77%) is comparable to the success rate of the formation of pairs in rhesus macaques, a far less socially complex situation [25,37,38]. Still, most of the procedure is designed based on the experience and insights of the behavioural specialists at BPRC. Until now, there were no possibilities to compare the outcomes and guidelines of the BPRC introduction procedure to the procedures used at other facilities. Therefore, we call for more studies describing and analysing male introduction procedures. Combining information from more institutes all over the world is important to find the best way to manage male introductions in captivity. At the moment, the BPRC introduction procedure can be used to optimize the management of captive primate groups at other facilities.

The introduction procedure described in this paper focusses on the introduction of a new male into rhesus macaque groups in the breeding colony of the BPRC. The BPRC houses many different rhesus macaque groups, providing the luxurious position wherein a new breeding male can be carefully matched with a new group in advance. However, even when the male cannot be matched to the group, which may be the case in zoo-housed primates, the BPRC introduction procedure can still be followed. The risks associated with male introductions may be larger in this case, as the selected male may not meet all the criteria used to select breeding males at the BPRC. Moreover, the BPRC introduction procedure may also be used to introduce new males into non-naturalistic primate groups, such as peer groups. The risks of female coalitions will be lower in peer groups, as there will be no matrilines that naturally support each other in the group [36,48,49,50] and the chances of female socially defeating a new male will be lower. However, females may still prevent successful introductions [28]. Thus, male introductions into non-naturalistic primate groups should also be conducted carefully. 

Finally, the introduction procedure described in this paper can also be used to introduce new males into existing social groups in other primate species. The BPRC uses the same procedure to introduce new males in rhesus macaques and long-tailed macaques. Long-tailed macaque male introductions may be expected to be more challenging due to the lack of seasonal breeding, leading to an increased likelihood of groups containing pregnant or lactating females throughout the year. Although the presence of pregnant or lactating females decreases male introduction success in rhesus macaques, BPRC has had 100% success with long-tailed macaque group formations (unpublished data). Species differences in introduction success may be due to the difference in how strict the female dominance hierarchy is, where long-tailed macaque hierarchies are more relaxed than rhesus [56]. These findings indicate that, although success rates may differ, the same procedure is successful for introducing males in captive groups of females of different primate species.

In summary, the BPRC introduction guidelines are widely applicable, and can be used by any facility housing captive primates, including zoos, rescue or rehabilitation centres, and research facilities. Thereby, the risk associated with male introductions and the chances of unsuccessful male introductions can be minimized, while enhancing the management of captive primate groups and primate welfare.

## Figures and Tables

**Figure 1 animals-11-00545-f001:**
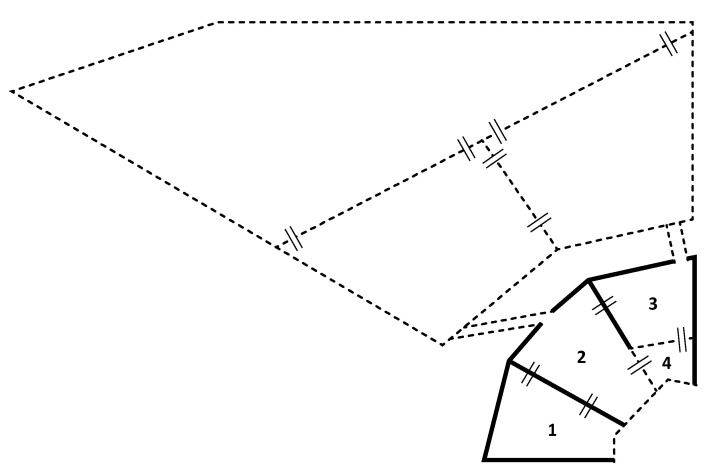
A map representing the inside and outside enclosure of the macaque housing facilities at the Biomedical Primate Research Centre (BPRC). The numbers indicate different inside compartments. Concrete walls are indicated by thick black lines, the dotted lines indicate wire mesh, and the thin black lines indicate sliding doors that connect the different compartments. 1: initial housing of new male: olfactory, auditory and limited visual contact; 2: housing of the female group; 3: housing of new male during familiarization phase: visual interactions only; 4: housing of new male during familiarization phase: visual contact and physical interaction through wire mesh. Various steps are described in detail under 2.3 and 2.4.

**Figure 2 animals-11-00545-f002:**
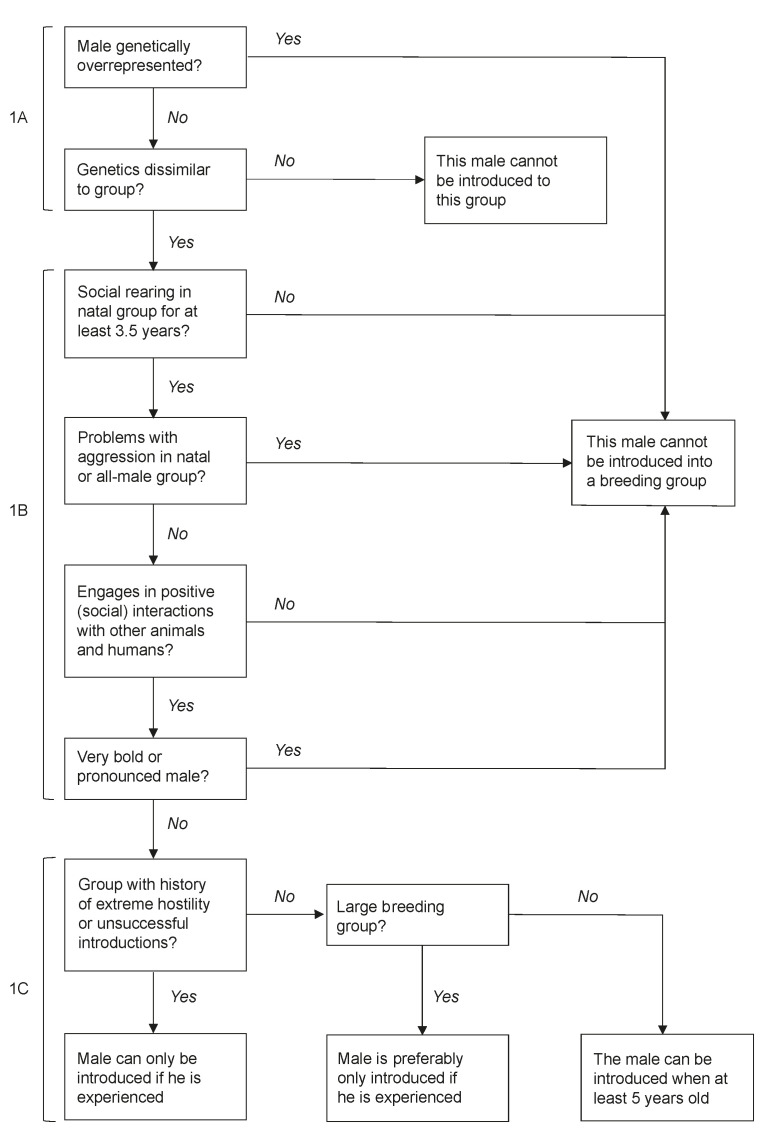
Step 1: The stepwise assessment to determine whether a male can be introduced into a particular group, concerning the new male’s genetic (1A), his social behaviour (1B), and body condition (1C).

**Figure 3 animals-11-00545-f003:**
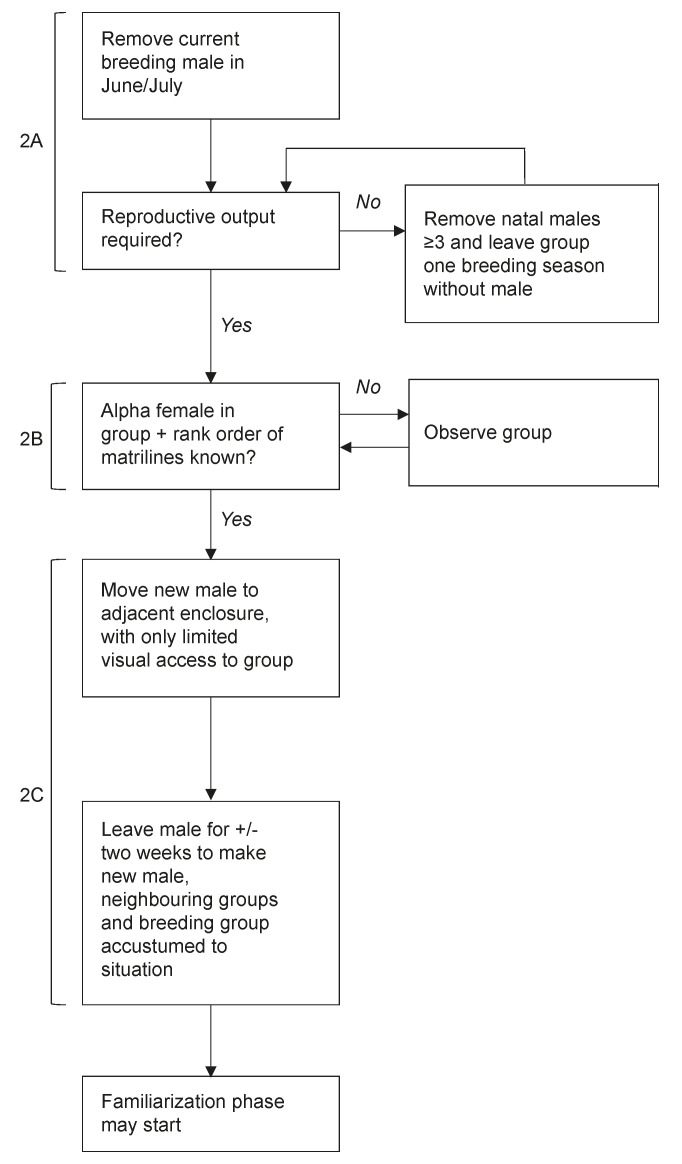
Step 2: The stepwise preparation of a male introduction, concerning the removal of the previous breeding male and sub-adult natal males (2A), observing the group (2B), and moving the new male (2C).

**Figure 4 animals-11-00545-f004:**
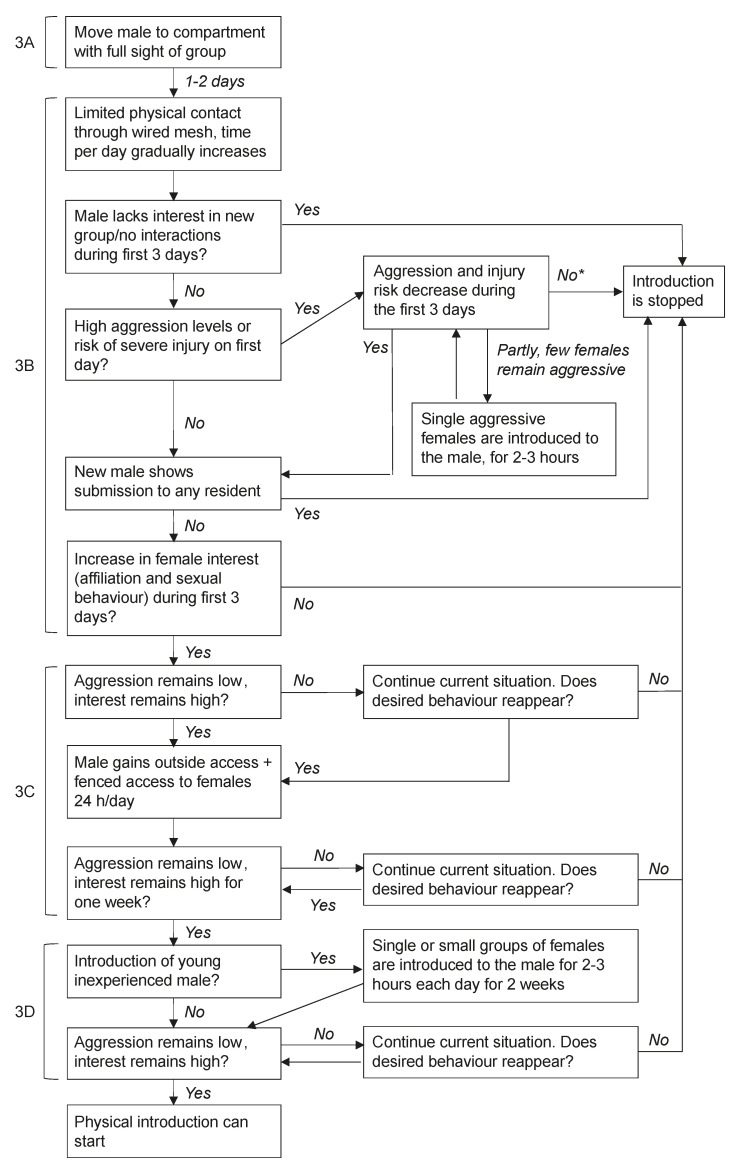
Step 3: The steps used to familiarize a new male and a breeding group, concerning providing full visual contact (3A), the first days of physical contact through wire mesh (3B), introducing individual females to inexperienced males (3C), and providing the male access to an outside compartment with limited supervision (3D). * When the introduction cannot continue in the group because of one or a few highly aggressive females who have a history of extreme hostility towards new males, an alternative strategy could be to remove the aggressive female(s) and continue the introduction.

**Figure 5 animals-11-00545-f005:**
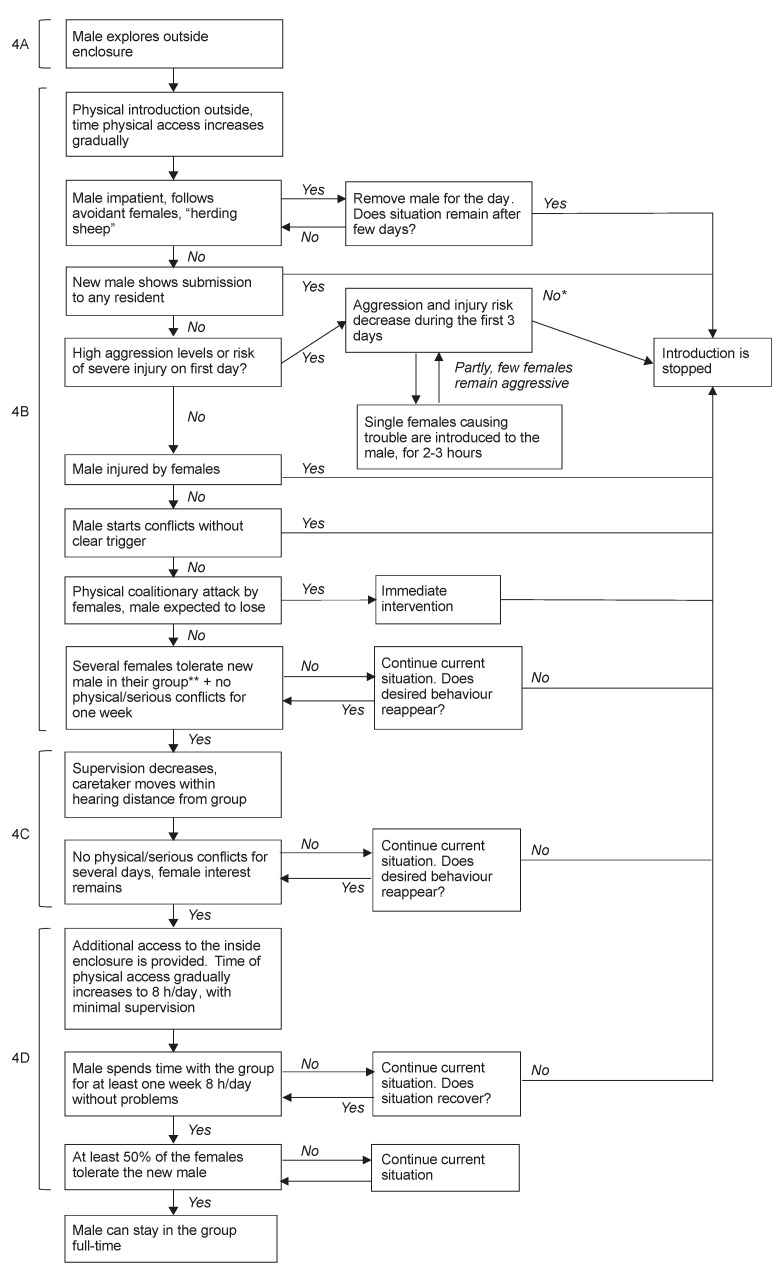
Step 4: The stepwise physical introduction of a new male into a breeding group, concerning the male exploring the outside enclosure (4A), the first days of physical interactions outside (4B), decreasing direct supervision (4C), and providing additional access to the inside enclosure (4D). * When the introduction cannot continue in the group because of one or a few highly aggressive females who have a history of extreme hostility towards new males, an alternative strategy could be to remove the aggressive female(s) and continue the introduction. ** for definition see [52].

## Data Availability

The data presented in this study are available on request from the corresponding author. The data are not publicly available because they are an integrated part of the general electronic database of the BPRC animal colonies.

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
