# Peer review of "A Stepwise Male Introduction Procedure to Prevent Inbreeding in Naturalistic Macaque Breeding Groups"

_animals, 2021, doi:10.3390/ani11020545_

Round 1

Reviewer 1 Report

The manuscript is a nice read and should be a strong contribution to Animals, I appreciated the authors thorough responses and edits to the previous review. There are still  frequent grammatical and sentence structure errors as well as some minor formatting errors that need addressing. The manuscript will likely benefit from a thorough editorial review but content wise is ready to go for publication.

Some of the things I caught though admittedly did not document all grammar/writing concerns as this is more editorial than content review. Additionally include two minor questions about content:

Line 29: Second half of sentence starting with “while” reads out of place. I think this part can be removed and authors’ points are still clear and flow of logic improves. Also uses “increase” with welfare, you do not increase welfare, but you can improve, optimize, enhance, etc. welfare.

Line 55: replace “such as most primates” with “particularly primates”? Reads oddly in current format.

Line 56-59: Bit of a run on sentence and the overall point is a bit scattered. Maybe break this into two sentences or one sentence with one point of emphasis that ties more strongly into the subsequent sentence?

Line 62: replace “social behaviour” with “social interaction” or “engagement in social behavior”?

Line 93: replace “a good” with “an ideal”?

Line 103: nice inclusion of data but I think an additional sentence explicitly tying in how this fits here would help drive the point home.

Line 130: migration is misspelled

Line 130: Suggest removing “as good as possible” given the sentence includes “mimic” which implies its not a perfect replication. Also mimic is spelled “mimick”, not sure if US vs Euro English difference or misspelling in general. Not sure of journals English preferences?

Line 188: add “at” before “least”

Line 198: what is “prime” body condition?

Line 202: replace “,” with “)”

Line 204: What do you mean by “strong”?

Line 223: Citation in incorrect format

Figure 3: 3rd box has unformatted symbols in it which are unclear?

Author Response

The manuscript is a nice read and should be a strong contribution to Animals, I appreciated the authors thorough responses and edits to the previous review. There are still  frequent grammatical and sentence structure errors as well as some minor formatting errors that need addressing. The manuscript will likely benefit from a thorough editorial review but content wise is ready to go for publication.

Thank you for your nice and positive comment. Please find below the answers to your questions/remarks (in bold/italic)

Line 29: Second half of sentence starting with “while” reads out of place. I think this part can be removed and authors’ points are still clear and flow of logic improves. Also uses “increase” with welfare, you do not increase welfare, but you can improve, optimize, enhance, etc. welfare.

Line 29: last part of the sentence has been removed.

Line 55: replace “such as most primates” with “particularly primates”? Reads oddly in current format.

Replaced (line 57)

Line 56-59: Bit of a run on sentence and the overall point is a bit scattered. Maybe break this into two sentences or one sentence with one point of emphasis that ties more strongly into the subsequent sentence?

The sentence has been broken into two sentences to tie it more strongly into the next sentence (line 58-60)

Line 62: replace “social behaviour” with “social interaction” or “engagement in social behavior”?

Replaced "behaviour" by "interaction" (line 62)

Line 93: replace “a good” with “an ideal”?

Replaced "a good" by "an ideal" (line 96)

Line 103: nice inclusion of data but I think an additional sentence explicitly tying in how this fits here would help drive the point home.

This paragraph describes the results of male introductions at the BPRC breeding groups, for pair-housed animals and for animals in the wild and show comparable (pair-housed) to sometimes less successful introductions (wild). This ties the different data together. 

Line 130: migration is misspelled

Corrected

Line 130: Suggest removing “as good as possible” given the sentence includes “mimic” which implies its not a perfect replication. Also mimic is spelled “mimick”, not sure if US vs Euro English difference or misspelling in general. Not sure of journals English preferences?

"as good as possible" has been removed. We now use "mimic" instead of "mimick"

Line 188: add “at” before “least”

Done

Line 198: what is “prime” body condition?

We changed this : " mature to full adult male". Also the last sentence of this paragraph has been moved to this part of the paragrapg to make it more logic.

Line 202: replace “,” with “)”

Done

Line 204: What do you mean by “strong”?

We have removed "strong"

Line 223: Citation in incorrect format

Corrected

Figure 3: 3rd box has unformatted symbols in it which are unclear?

Our apologies. We have corrected this.

Reviewer 2 Report

The authors have addressed my initial concerns quite well and I have only a few editorial suggestions:

  1. 130 “migration”, “as closely as possible”
  2. 222 There is a complete citation here that is not in the references.

Figure 3, Step 2 A There are a lot of symbols in the box that begins “Remove natal males”

  1. 277, 279 and many other places—the mesh is referred to as “wired mesh”, “wire mesh” and “fencing” which I assume all mean the same thing. I suggest “wire mesh” be used consistently throughout the paper.

Figure 4 Step 3 A “with full sight of group”

Figure 4, Step 3 C add a “yes” to the arrow from “continue current situation” to “male gains outside access”

Author Response

The authors have addressed my initial concerns quite well and I have only a few editorial suggestions:

Thank you very much for your nice comment and your suggestions

130 “migration”, “as closely as possible”

“as closely as possible”  has been added

222 There is a complete citation here that is not in the references.

This has been corrected (reference 47 in revised manuscript).

Figure 3, Step 2 A There are a lot of symbols in the box that begins “Remove natal males”

Our apologies. In our copy this was OK but aparantly oemthing went wrong with sending (all reviewers had the same remark). We have added a corrected Figure.

277, 279 and many other places—the mesh is referred to as “wired mesh”, “wire mesh” and “fencing” which I assume all mean the same thing. I suggest “wire mesh” be used consistently throughout the paper.

“Wire mesh” is now used throughout the manuscript.

Figure 4 Step 3 A “with full sight of group”

This has been corrected

Figure 4, Step 3 C add a “yes” to the arrow from “continue current situation” to “male gains outside access”

This has been done

Reviewer 3 Report

I am pleased with the revisions the authors have made to the manuscript. I have identified a few more clarifications in the attached document that should be addressed before publication.

Author Response

Thank you for your review and the suggestions. Please find below our response to your questions, remarks and suggestions.

56                    Please change “is impaired” to “can be impaired

Changed as suggested

66                    It seems odd to use the term migrations in relation to captive primate social groups as current cage design prevents this. I suggest to rephrase this sentence as "The composition and migration patterns within these groups is managed so that they closely resemble those in nature ... " 

This has been adapted accordingly

74                    2,5 should be adjusted to 2.5 for English text. 

Corrected

99-101             Check the parentheses structure in this sentence and keep the structure of how "i.e.," is written (within or without parentheses, with or without a comma) consistent. 

We have made this consistent (within parentheses and with comma).

102                  A dash is more appropriate than a semicolon here. 

Replaced by “comma”

150-151             Why did this introduction go on so long? Is there not a introduction termination criteria if no or limited interactions are occurring? Could you please add a brief explanation if appropriate? 

We have changed this slightly and added an explanantion. “One successful introduction took 357 days [25]. Usually, such long introduction period is only carried out in exceptional cases, in this case because the man was genetically important for this breeding group”

176                  Is this the first time that MHC is written in the manuscript? If so, please write out the whole phrase and then indicate the abbreviation to be used in the manuscript. 

We have included “major histocompatibility complex”

Figure 3           It appears that the text did not export well in this figure. Please check formatting. 

Our apologies. Apparently this went wrong with sending. Has been corrected.

260                  I suggest to use "cage relocation" instead of "moving" to distinguish from physical body movement. 

“Moving” has been replaced by “cage relocation”

397-399           The phrasing of this sentence is confusing. I suggest " ... by high levels of aggression, submissive behavior by females to the new male and mating, regardless of who initiates these behaviors; in contrast, bouts of affiliation (e.g., grooming) are rare." 

We have changed the sentence as suggested

416-417           Needs rephrasing. I suggest "In general, solo introductions can help decrease the hostility of highly aggressive females towards a new male." 

We have changed the sentence as suggested

469-471           Simplify these sentences. I suggest "Yet, male introductions are risky and may not always result in success, aspects which can be influenced by the procedure itself 

[citations]." 

We have changed the sentence as suggested

475                  "lead" should be replaced with "led" as the sentence is in the past tense. 

Corrected

539-540           I suggest rephrasing as" ... may be due to the difference in how strict the female dominance hierarchy is, where long-tailed macaque hierarchies are more relaxed than rhesus." 

We have changed the sentence as suggested

542                  I suggest replacing "successful in" with "successful for" and specify that the males are introduced to captive social groups of females to bring the paragraph to a tidier close.

We have included your suggestions

This manuscript is a resubmission of an earlier submission. The following is a list of the peer review reports and author responses from that submission.

Round 1

Reviewer 1 Report

This manuscript reviews the procedures used by a biomedical laboratory to introduce male macaques into breeding groups.

As a scientist working in an animal husbandry research field I appreciate the authors goal of sharing their husbandry methodologies with a larger audience. This is not done enough and needs to be a point of emphasis for professionals in the field, especially when rather larger undertakings, such as those described here, are the focus.

That being said I do not think this manuscript is an appropriate fit for Animals as it has been submitted. It is submitted in a format of a research article, though no formal study has taken place. I am not familiar with the format, but Animals does accept a "communication" submission, which may be appropriate, but feedback from the editors should be taken before doing so. If such an alternative format here is not sufficient, other journals such as Zoo Biology and Zoo and Aquarium Research do have a husbandry report format, which would be a more more appropriate fit.

When resubmitting, whether here or elsewhere, I also feel this article would benefit from a few major/moderate edits.

  • Have a native English speaker review for grammar. Overall this is a well written manuscript but some grammatical aspects stand out and should be addressed.
  • Overall the manuscript is a bit long. Often I felt that concepts were being described in "detail" though there was not much informative information being shared. I feel the authors can streamline a large amount of the content here.
  • Aspects of the methods should be have more context supporting them. Many statements are made without sufficient background to understand the applied work/value that went into it. For example (but not limited to):
    • Line 100: "approximately 77% of the male introductions are successful." What defines success? How can you definition of success be shared here and have applied value to other organizations?
    • Line 149: "It may take up to a year before a new breeding male is introduced." What is the average? Does length of time without a male play into success? Opportunities for more data inclusion here would be informative.
    • Section starting on Line 170: genetic relatedness is discussed. How is that assessed/measured and used here?
    • Line 187: "inappropriate aggression towards other animals." How is this defined, monitored and tracked? How are these data then summarized and assessed to guide decisions?
    • Line 189; "not become injured or injure other males." How are you defining injury? Macaques can be quite agonistic. How is wounding/injury monitoring, tracked and applied in this context?
    • Line 198: "introduced into smaller breeding groups." What defines small? Are there size recommendations here you can make? Are their data you can summarize around success in relation to group size?
  • This is completely minor, and nit picky, but it caught my attention. On line 120 the word "spacious" is used to describe the enclosures. I feel it is best to not use such descriptors when describing housing conditions, especially when they are accompanied by actual dimensions that seem to be rather on the smaller size.

Reviewer 2 Report

This is a paper with great relevance to animal welfare. It describes in great detail methods of introducing a novel male to a natural social group of monkeys of multiple matrilines in order to prevent inbreeding. This is valuable not only for those managing biomedical research colonies, but also for zoos and others working with captive primates. It is surprising that a similar paper has not previously been published, but I have not come across any. I know that many zoos practice something similar for introducing new animals, but am not aware of any detailed elaboration such as this paper provides. The authors do an excellent job of providing clear detailed decision trees for selecting a suitable male for introduction, habituating group members to the new arrival, engaging in carefully staged exposures and careful follow up during the physical introduction.

I have no substantive critique, but since many primates are patrilocal and others have different social structures (paired, family living), I am curious about how the authors think their methods would transfer to female introductions or with species with different social structure from macaques.

Minor points:

  1. 18 Italicize Latin binomial
  2. 57, some, but not all primates live in complex multi-male, multi-female groups. Macaque society structures are not universal across all primate species.
  3. 74 what is the mean or median age of puberty in captive macaques?
  4. 80 Captive introduction of novel males may be risky, but are there any good field data on how risky it is for wild macaques?
  5. 86 why are authors’ names and dates given here when numbering is the preferred style of the journal?
  6. 101 why is 70% repeated twice here? The PLoS paper by the same authors seems to indicate a smaller percentage of males staying in long term stable groups.
  7. 185 et seq. be sure that verbs are in the past tense here. There is currently a mix of past and present tense within the same sentence.
  8. 189 “have been involved …. injured”
  9. 326 “through” not “though”
  10. 377 “cause a restraint on”
  11. 410-11 there is no verb connecting the two “when” clauses.
  12. 418 “responds with high aggression” also in next line.
  13. 435 are “trigger” and “initiator” redundant? Just use one or the other.
  14. 442 “all females”

Figure 5 under 4 B second box “Impatient” not “unpatient”

  1. 495 “are a particular risk”

References are not consistent in terms of capitalizing initial title word versus most words in title. Should not Latin binomials be italicized in the references?

This is a paper with great relevance to animal welfare. It describes in great detail methods of introducing a novel male to a natural social group of monkeys of multiple matrilines in order to prevent inbreeding. This is valuable not only for those managing biomedical research colonies, but also for zoos and others working with captive primates. It is surprising that a similar paper has not previously been published, but I have not come across any. I know that many zoos practice something similar for introducing new animals, but am not aware of any detailed elaboration such as this paper provides. The authors do an excellent job of providing clear detailed decision trees for selecting a suitable male for introduction, habituating group members to the new arrival, engaging in carefully staged exposures and careful follow up during the physical introduction.

I have no substantive critique, but since many primates are patrilocal and others have different social structures (paired, family living), I am curious about how the authors think their methods would transfer to female introductions or with species with different social structure from macaques.

Minor points:

  1. 18 Italicize Latin binomial
  2. 57, some, but not all primates live in complex multi-male, multi-female groups. Macaque society structures are not universal across all primate species.
  3. 74 what is the mean or median age of puberty in captive macaques?
  4. 80 Captive introduction of novel males may be risky, but are there any good field data on how risky it is for wild macaques?
  5. 86 why are authors’ names and dates given here when numbering is the preferred style of the journal?
  6. 101 why is 70% repeated twice here? The PLoS paper by the same authors seems to indicate a smaller percentage of males staying in long term stable groups.
  7. 185 et seq. be sure that verbs are in the past tense here. There is currently a mix of past and present tense within the same sentence.
  8. 189 “have been involved …. injured”
  9. 326 “through” not “though”
  10. 377 “cause a restraint on”
  11. 410-11 there is no verb connecting the two “when” clauses.
  12. 418 “responds with high aggression” also in next line.
  13. 435 are “trigger” and “initiator” redundant? Just use one or the other.
  14. 442 “all females”

Figure 5 under 4 B second box “Impatient” not “unpatient”

  1. 495 “are a particular risk”

References are not consistent in terms of capitalizing initial title word versus most words in title. Should not Latin binomials be italicized in the references?

Reviewer 3 Report

This manuscript is an important contribution to the field of captive primate group management, describing how to conduct successful one-male, multi-female group introductions in macaque species. The authors have described their methodology well and in great detail. Furthermore, the methodology has been synthesized into flow diagrams nicely, a difficult task given that macaque group formations can differ greatly depending on the individuals and the composition of the group.

Generally, it would be helpful to include more data behind the methodology to provide context for other facilities. For example, how long are social groups kept together and how frequently do resident females receive a new adult male? How long to introductions generally take? How big is a typical/optimal group (and how many matrilines) at the time of group introduction? Data (e.g., range, mean) on how long each step typically takes would be useful to have alongside the flow chart figures (figures 3,4, and 5). Although this type of data/information may be present in a previous publication, it bears repeating/citing either in the introduction or in the methods for other behavioral management programs developing their own guidelines. Additionally, I recommend more specific or apparent definitions for successful/unsuccessful introduction, naturalistic/non-naturalistic group, stable/unstable group, high/low level of aggression, submission signals observed, males that are “genetically overrepresented” or “dissimilar” from resident females, etc. as they would be useful for other facilities to compare with their own definitions.

The manuscript would also benefit by having a native British English speaker read through it. I have provided suggestions and/or pointed out mistakes when possible in the lines below. One common clarification that can be made throughout the manuscript is to specify what “this” (usually occurring at the beginning of a sentence) is referring to (e.g., line 95, 228). Currently, the lack of specificity or reference requires the reader to remember or to go back and find the subject/topic in the previous sentence. Clarifying the subject/topic within these sentences will help with the flow of the article greatly.

Below I have included more specific comments by providing the line number(s) first, followed by the comment.

Specific comments:

19        Please remove the word “introductions” here.

20        I think it is important to emphasize here what the group structure consists of as the outcomes of introductions can vary dramatically based on group structure. Theses males are being introduced to a group of females and their offspring, not a group containing other adult males.

24        Please replace “in” with “during”.

32        Needs rewording. I suggest: "…may affect the success and influence the risks associated with group introductions.”

34        Please clarify the group structure here as one male, multi-female.

38        Replace “will provide” with “provides” and “on” with “as to”.

47-49   These sentences require revision. It may be useful to first clarify that animals are found in many different types of captive environments and why it is important for people (not just scientists) to promote animal welfare. Animals are also used in many basic research studies and are a major source of food. Welfare in these environments are also under intense scrutiny so they are important to mention.

54        Should be “assessing an animal’s quality..”

58-59   Do you mean that welfare is impaired when the opportunity to socialize is removed completely or when the scope/breadth of social behavior is limited due to captive management reasons? Please clarify.

62        Natural social behavior can also occur in pair-housing environments; however, the breadth of the behaviors exhibited is usually more limited. Please clarify.

66        How large? A range of typical group size would be helpful here for behavioral management readers. Also, I believe “mainly” should be moved between  “are” and “housed”.

63-68   Group composition and demography are redundant in the same sentences.

63-70   An example of “non-naturalistic conditions” of group housing at other institutions would be useful to demonstrate how the BPRC social group housing method is different and more naturalistic. Additionally, please be careful throughout the manuscript with equating the one male, multi-female group structure as the norm for macaque species. While the examples provided in lines 68-70 are good support for using the group structure at BPRC, these examples are likely exceptions and/or occur for a short period of time in the wild.

73-74   Please clarify that males are removed at BPRC and specify the age at which they are removed. Readers not familiar with macaques may not be aware of what age this occurs at and definitions of puberty likely vary between facilities.

75        Please replace “remains” with “remain”

82        Please remove either “In addition” or “also” to reduce redundancy. I suggest: “Females and their offspring are also at risk…”

84        “both” is unnecessary here.

86-88   I suggest combining these sentences somehow. For example, “Due to these risks, it is important to consider all management strategies to facilitate social group formation for the purposes of preventing inbreeding…”

88-98   How common is the method of peer-rearing groups by taking them from their existing social group amongst research facilities? To my knowledge, peer-rearing groups become a necessity when animals are born to pair-housed mothers and must be removed due to the limitations of the pair-housing environment; therefore, peer-rearing juveniles with an experienced adult male and female pair is the next best thing as it is difficult to integrate juveniles into an existing social group. Additionally, what happens to the young males that are removed from BPRC groups? Are they transferred to another social group?

95        Please clarify what “This” means.

101      What is the definition of “successful”?

113      Please replace “cam” with “can”.

127      To reduce wordiness, I suggest: ”different compartments, either separated by mesh or concrete.”

130      Please replace “ranked” with “ranking” to be consistent with tense.

134-138           By thin black lines for the sliding doors, do you mean those that occur as pairs next to each other? Please clarify in the figure legend. Also, it is not immediately obvious that there is a difference between the grey and black lines. It may be helpful to thicken the lines in general or add in dashed lines to indicate there the wire is.

148-149           Is there a threshold or criteria (e.g., eldest daughter reached breeding age) at which the risk of inbreeding is determined to be high? Please specify.

149-150           Why does it take a year to introduce a new male? Are there not enough males available or does the introduction process take a year to conduct?

152      Is alpha status is a BPRC policy for introductions?

154-155 Please replace “all crucial to optimize” with “all crucial for optimizing”

159-160           Is the duration of introduction phase (based on behavior) determined beforehand and it is the same for every introduction? The term “exact duration” suggests this, but it seems like the duration of each phase is dependent on how well the caretaker thinks the introduction is going based on the behaviors observed. Please clarify.

171-179           What is the criteria for what makes a male “genetically overrepresented” or “dissimilar”? Currently this information is vague and would be useful for other breeding group management programs.

189      So a male cannot become a breeding male if he receives or give any injury in another group? Could you please specify why not? Also, what is the reason for the former? If he is being introduced to a group of females, the risk of injury should be quite low. As a note, this seems like very strict criteria and may not be feasible for other facilities to implement.

191-192           What does it mean to be “too pronounced”? Also, why are bold males unsuccessful?

170-205           Generally, this section would benefit with more specific data behind the reasoning for selecting particular males rather than only referring to personal communications if possible.

209-211           These sentences can be combined.

228      Please clarify what “This” refers to. For example, “This practice…”.

232      Please replace “shows” with “show” or “indicate”

235-37 These lines are unnecessary as they are a repetition of what was mentioned earlier in the paragraph.

Figure 3, panel 2C      “accostumed” is incorrect spelling; please replace with “accustomed”

245      Please rephrase sentence. I suggest “Moreover, it is necessary to determine the rank order of group matrilines and which females belong to those matrilines.”

246-247           Not only predict, but help identify the culprits of escalating aggression quickly, correct?

249      I would replace “goes wrong” with “is unsuccessful” or another term.

258-60             These sentences can be combined.

260      replaces “will allow” with “allows”

258-271           I think it is important to state how long the solitary housing period is earlier in this paragraph. Is two weeks the minimum or maximum of solitary housing duration? Or do the caretakers rely on behavioral signals to indicate that the male has adapted to the new environment? E.g., relaxed posture, less pacing, etc. What is the specific criteria for moving onto the next step?

262      Please clarify what “This” referring to.

265      Please clarify what “This” referring to.

275      Space needed in between text and citation.

298      What signals of submission are considered when physical access is not possible? Please specify.

323      Please specify “This strategy” instead of just “This”.

324      Please clarify what is meant by “this”.

325-327           Please correct the tense of the sentence to be the same as the rest of the paragraph.

327      This sentences needs a transition phrase such as “During the introduction,…”

332      Please check the phrase “in its turn”; “in turn” may be the correct phrase.

333      I suggest to replace “will be better able to” with “can” and add “more easily if they should occur” at the end of the sentence.

335      “introduction stage” may be a better suited word than “situation” to be more specific.

338      Is the composition of the sub-groups random or do the caretakers introduced specific matrilines, animals, etc. at a time?

344      Please consider “help an inexperience male practice” instead of “help an inexperience male to practice”

347      The final step meaning the introductions to sub-groups of females? Please clarify.

350      “slowed down” meaning spend more time conducting physical introductions with sub-groups for females? Please clarify.

351      In the case that aggression does not decrease? Please clarify.

Figure 4, panel 3A      Please correct spelling of “comparment” to “compartment”.

372-380           This text is better suited following the first sentence of the next paragraph as it describes the reasoning for locking the animals outside in the next introduction step.

376      How do the caretakers stop or interfere with an introduction in case it goes sour? Are the animals well-trained to separate? May be informative for other behavioral management programs.

377      I suggest to replace “cause a restrain on” with “limit” or “repress”.

398      It seems that the sentence is incomplete. Interact with the group through what? Through the mesh?

399      Please replace “continuous” with “continues”.

400-402           Please rephrase sentence. Perhaps, “The beginning of physical introductions are characterized by…, while affiliation rarely occurs.”

404      Do you mean that the male may threaten females to confirm his dominant position? Or that the females are threatening him. Please clarify.

408      Please give an example of what types of aggression are considered extreme or define here. Definitions may vary in other facilities. Additionally, is there a certain level or types of aggression that are allowed without the caretakers interfering?

410-411           Please revise sentence. Removing “When” from the beginning of the sentence may be helpful.

417      Within three days of physical introduction? Please clarify.

420      What do you mean by “this”? The solo introduction to the male? Please specify.

442      Please revise “all female” to “all females” if appropriate.

443      Please consider replacing “grow” with “develop”.

Figure 5, panel 4B      “avoidant” is a more suitable word than “avoiding” in the second box on the left. In the box labeled “immediate intervetion”, correct the spelling of “intervetion” to “intervention”.

Figure 5, panel 4C      In the first box, “caretaker moves within hearing distance” is more suitable than the current phrasing.

457      “proving” is incorrect; I suggest “provision of”.

469      Please replace “payed” with “paid”.

478      I suggest “…a successful procedure for introducing…” to replace the current phrasing.

480      Please add in what the success rate is and repeat the definition of what successful means in the context of this paper.

481      I suggest “Sharing this procedure will help improve male introduction management at other facilities…” to replace the current phrasing.

484      I suggest “Particular attention should be given to the male’s ability to win fights against female coalitions, as it is important…” to replace the current phrasing.

486-487           So males are not re-tried in another formation of novel females due to the previous unsuccessful attempts at the BPRC? Please clarify and rephrase this sentence.

487      I suggest “These observations are in line with the winner-loser…” to rephrase the beginning of the sentence.

 492-497          Please repeat who “they” refers to within these sentences as it is difficult to follow that “they” is referring to the caretakers/ethologist after a few sentences.

498      I suggest rephrasing this line as “..introduction, whereas introductions that are too quick can have…”

501-502           I suggest rephrasing as “and have identified potential behaviours signaling whether the introduction is progressing well or not,…”

504      I suggest rephrasing as “…there is no perfect way to introduce…”

501 & 508       It would be good to remind the reader that these are macaque male introductions at the beginning of these paragraphs and elsewhere given that the paper is submitted to the journal “Animals”.

523-525           This sentence is confusing. Please clarify.

526      I believe “all the criteria used to select breeding…” is the correct phrasing here.

528      Can you please clarify what is meant by non-naturalistic primate groups? Peer-reared groups? All male groups (which can occur in the wild, but are not as common)? Female only groups?

528-532           These sentences can be combined and the text reduced.

533      A separate paragraph would be beneficial before the sentence beginning with “Finally”.

535-541           Please rearrange the content of this text to have a better flow. Here is my suggestion: “Long-tailed macaque male introductions may be expected to be more challenging due to the lack of seasonal breeding, leading to an increased likelihood of groups containing pregnant or lactating females throughout the year. Although the presence of pregnant or lactating females decreases male introduction success in rhesus macaques, BPRC has had 100% success with long-tailed macaque group formations. This success rate shows…”

540      Do you have an idea why male introductions are more successful with long-tails? It may be useful say so here.

Other comments:

Why not use contraception to prevent males from breeding with their daughters until many have reached adulthood? Wouldn’t this allow for social groups to remain together for longer if breeding is not required? Is this not the policy at BPRC currently?

Is there a reason for introducing only one adult male at a time? This aspect has been ignored in the manuscript and should be addressed either in the introduction or methods. Although there may be higher risk of aggression/injury between the males, wouldn’t more males diversify the gene pool within the breeding group and aid in policing female aggression? Additional males would also help buffer the physiological demands of policing and mating by spreading out the “work”. Dramatic weight loss during the mating season can be a health concern for (rhesus) males, particularly when there is only one in a group of females. Furthermore, adding in other males would also create more social housing opportunities for males in general and give young, unrelated males the opportunity to gain social experience and learn from an experienced role model.
